# Mid-Phase Hyperfluorescent Plaques Seen on Indocyanine Green Angiography in Patients with Central Serous Chorioretinopathy

**DOI:** 10.3390/jcm10194525

**Published:** 2021-09-30

**Authors:** Elodie Bousquet, Julien Provost, Marta Zola, Richard F. Spaide, Chadi Mehanna, Francine Behar-Cohen

**Affiliations:** 1Department of Ophthalmology, Ophtalmopôle, Hôpital Cochin, Assistance Publique-Hôpitaux de Paris, AP-HP, Université de Paris, 75014 Paris, France; marta.zola9@gmail.com (M.Z.); chadimehanna@gmail.com (C.M.); 2Centre de Recherche des Cordeliers, Sorbonne Université, Université de Paris, Inserm, from Physiopathology of Retinal Diseases to Clinical Advances, 75006 Paris, France; 3Vitreous, Macula, Retina Consultants, New York, NY 10022, USA; rick.spaide@gmail.com; 4Department of Biostatistics, Hôpital Necker-Enfants Malades, AP-HP, 75015 Paris, France; 5Department of Ophthalmology, American Hospital of Paris, 92200 Neuilly-sur-Seine, France

**Keywords:** central serous chorioretinopathy, indocyanine green angiography, choroidal hyperpermeability

## Abstract

(1) Indocyanine green angiography (ICG-A) shows the presence of mid-phase hyperfluorescent area in central serous chorioretinopathy (CSCR). However, their exact meaning remains uncertain. (2) The clinical and multimodal imaging findings of 100 patients (133 eyes) with CSCR, including the enhanced-depth-imaging OCT (EDI-OCT), blue-light fundus autofluorescence (BAF), fluorescein and indocyanine green angiography (FA and ICG-A) findings were reviewed. Mid-phase hyperfluorescent plaques (MPHP) were defined as fairly well circumscribed hyperfluorescent regions during the midphase of the ICG-A. The association between MPHP and other clinical/imaging parameters was assessed using a multiple logistic regression analysis. (3) MPHP were detected in 59.4% of eyes with CSCR. The chronic form of the disease, the presence of irregular pigment epithelium detachments (PED) and the retinal pigment epithelium (RPE) changes seen on FA were associated with the presence of MPHP in the multivariate analysis (*p* = 0.015; *p* = 0.018 and *p* = 0.002; respectively). OCT showed RPE bulges or PED in 98.7% of areas with MPHP and BAF showed changes in 57.3% of areas with MPHP. (4) MPHP were associated with a chronic form of CSCR and colocated with PED or RPE bulges. MPHP should be recognized as a sign of early RPE dysfunction before it is detected with BAF.

## 1. Introduction

Central serous chorioretinopathy (CSCR) is a chorioretinal disease characterized by the presence of serous retinal detachments (SRD) associated with retinal pigment epithelium (RPE) detachments (PED) and an increased choroidal thickness, occurring more commonly in middle-aged men [1]. One of the characteristics of CSCR is its frequent association with dilated choroidal veins in the Haller layer beneath a thinning of the choriocapillaris and the Sattler’s layer [2], likely to suggest a vein overload.

To date, several risk factors for CSCR have been identified. The most consistent risk factor is an exposure to glucocorticoids [3]. Several other predisposing factors or contributing factors have been identified as: psychological stress, anxiety and maladaptive personality [4,5], hypertension and cardiovascular disease, sleep disturbance and shift work [6], allergic disorders, pregnancy, peptic ulcer disease and also genetic risk factors [7].

Indocyanine green angiography (ICG-A) remains the gold standard imaging method to identify the dynamic choroidal vascular changes occurring in CSCR. Early-phase ICG-A shows a delayed choroidal filling and a dilation of choroidal veins, while geographic areas of hyperfluorescence are visible in the mid-phases and fade away in the late phases [8] (Appendix A) [8,9]. They are not necessarily associated with observable pathologic manifestations, remain stable over years, independent from subretinal fluid and have not been observed only in the affected but also in the fellow unaffected eye of CSCR patients [8,10]. While it is widely accepted that hyperfluorescent areas during ICG-A result from dye leakage from leaky vessels, such as dilated veins or the choriocapillaris, this hyperpermeability has more than one clinically evident manifestation. Simple hyperpermeability may lead to increased dye transudation into the choroid. These areas have poorly defined outer borders, expand in size during the course of the angiogram, and produce faint hyperfluorescent regions that silhouette the larger choroidal vessels in the later phases of the angiogram [8]. But, there are other eyes that show more well-defined areas in which the underlying choroidal details are obscured. The regions appear few minutes after ICG injection and increase in intensity until 7 to 10 min and do not show expansion during the course of the angiogram. They have well defined borders and disappear at the very late phase of the angiogram (>20 min). Interestingly, the angiographic dynamics of these mid-phase hyperfluorescent plaques (MPHP) parallels the fate of ICG from the choroid into the RPE observed in post mortem eyes from monkeys that were enucleated at several intervals after ICG was injected intravenously [11]. At 15 min after injection, ICG was mostly found in the RPE cells and in the Bruch membrane, suggestive of an active transport of ICG from the choroidal endothelial cells to the RPE, which was further confirmed by in vitro study showing that RPE cells internalize ICG through active mechanisms [12].

In this context, the aim of this study was to investigate the multimodal imaging findings and clinical parameters associated with CSCR eyes and to evaluate the multimodal imaging findings of MPHP using enhanced-depth-imaging OCT (EDI-OCT), blue-light fundus autofluorescence (BAF), fluorescein angiography (FA) and ICG-A. The main conclusion of the study is that MPHP could be the clinical manifestation of RPE dysfunction and not only of vascular permeability.

## 2. Materials and Methods

### 2.1. Ethics Statement

The study was approved by the Ethics Committee of the French Society of Ophthalmology (IRB 00008855 Société Française d’Ophtalmologie IRB#1). The study adhered to the tenets of the Declaration of Helsinki (1964). Signed informed consent was obtained from all subjects.

### 2.2. Study Design

This was a retrospective study conducted in the departments of ophthalmology of Hotel Dieu and Cochin Hospitals, Paris, between 2012 and 2020.

### 2.3. Study Patients

The medical records and imaging findings of consecutive patients with central serous chorioretinopathy (CSCR) imaged with indocyanine green angiography (ICG-A) were retrospectively reviewed. CSCR was diagnosed by two retinal experts (EB, FBC). Exclusion criteria were patients with drusen within 30 degrees, dome-shaped macula, vitreomacular traction syndrome, any history of uveitis, other retinal diseases, or poor image quality.

### 2.4. Study Protocol

The medical data collected included age, gender, history of corticosteroid intake, shift work, previous CSCR treatments and the best-corrected visual acuity (BCVA) converted into LogMAR scale. The clinical form of the disease was also recorded. Chronic CSCR was defined by the presence of a persistent serous retinal detachment for at least 6 months. In other cases, CSCR was classified as acute/recurrent.

The multimodal imaging data included spectral-domain OCT with enhanced-depth-imaging mode (Heidelberg Spectralis, Heidelberg, Germany), blue fundus autofluorescence (BAF), fluorescein and indocyanine green angiography (FA and ICG-A, Spectralis).

### 2.5. Image Analyses

The choroidal thickness was manually measured on the EDI horizontal B-scans passing through the fovea as previously described [13].

The presence of pigment epithelium detachment (PED) was assessed on the SD-OCT B scan. A bulge of the RPE was defined as a protrusion of the RPE without the visualization of the Bruch’s membrane as previously described [14]. The PED was defined by an elevation of the RPE with a distinct visualization of the Bruch membrane. The form of PED was also analysed and classified as dome-shaped or irregular PED defined by an irregular elevation of the RPE [15].

The presence of hyper or hypo autofluorescent area and gravitational tracks was assessed on BAF. The presence of CSCR focal leak with a smokestack and ink-blot pattern was assessed on FA. The presence of dilated choroidal veins (pachyvessels) was assessed on early-phase ICG-A, and the presence of mid-phase hyperfluorescent plaques (MPHP) on ICG-A was recorded.

All imaging analyzes were performed by two trained retinal specialists (JP and MZ). Disagreements were resolved by a third retinal specialist (FBC).

### 2.6. Statistical Analysis

Statistical analyzes were performed using XLstat software (version 2020; Addinsoft Paris, France), and R version 3.6.3 (The R Foundation for Statistical Computing). Descriptive data are presented as the mean ± standard deviation for quantitative variables and as counts and percentages for categorical variables. A univariate logistic regression was performed to assess the association between the outcome variable MPHP and the independent variables. Correlations between both eyes were eliminated using the marginal generalized estimating equations (GEE) model. Factors showing significant associations in the univariate analysis (*p* < 0.10) were included in the multivariate regression model using a stepwise backward variable elimination. Adjusted odds ratios and their 95% confidence intervals were calculated for the factors included in the multivariate logistic regression models. All *p* values were 2-sided and *p* values ≤ 0.05 were considered statistically significant.

## 3. Results

### 3.1. Patient Characteristics

The study cohort included 133 eyes of 100 patients. Patients’ demographics, clinical and imaging data are summarized in Table 1. Patients mean age was 47.6 (±10.2) years. Ninety-one patients (91%) were men. A previous corticosteroid intake was reported in 42 patients (47.2%). In 83 eyes (62.4%), CSCR was chronic. A unilateral form of CSCR was observed in 68 patients while 32 patients had both eyes affected. The mean (±SD) subfoveal choroidal thickness was 461.3 µm (±104.5) µm. PED and/or RPE bulges were detected in 117 eyes (88.7%). Gravitational tracks were reported in 36 eyes (27.1%). On ICG-A, mid-phase hyperfluorescent plaques (MPHP) were seen in 79 eyes (59.4%, Figure 1) and they were multifocal in 96% of the cases.

### 3.2. Clinical and Imaging Findings of Patients with and without Mid-Phase Hyperfluorescent Plaques

The comparison of the characteristics of patients with and without MPHP is summarized in Table 2. The rate of MPHP was significantly higher in eyes with a chronic form of the disease (66 eyes (83.5%) versus 17 eyes (31.5%); *p* < 0.001). In addition, the rate of irregular PED was higher in eyes with MPHP (40 eyes (51%) versus 10 eyes (18.5%); *p* < 0.001). On BAF, more eyes with MPHP had BAF abnormalities and gravitational tracks (*p* < 0.001 and *p* = 0.001, respectively). On early-phase ICG-A, a foveal dilated vein was more frequently visualized in eyes with MPHP (*p* = 0.001).

### 3.3. Factors Associated with the Presence of Mid-Phase Hyperfluorescent Plaques

In the multiple logistic and LASSO regression analysis, a chronic form of the disease, the presence of irregular PED, the RPE changes seen on FA and a foveal dilated vein seen on ICG-A were associated with MPHP in CSCR patients (*p* = 0.015; *p* = 0.018; *p* = 0.002 and *p* = 0.036; respectively Table 3).

### 3.4. Multimodal Imaging Analysis of the Areas with Mid-Phase Hyperfluorescent Plaques

A total of 249 MPHP were detected in 79 eyes and analyzed by multimodal imaging (Table 4). On early phase ICG-A, available for 132 MPHP, a delayed choroidal filling was detected in 77.3% of areas with MPHP. Dilated choroidal veins were observed in 42.8% of areas with MPHP (data available for 243 MPHP). On FA, a focal leak was detected in 25 eyes with MPHP and it was located in areas with MPHP in 20 eyes (80%). BAF abnormalities were detected in the area of MPHP in 57.3% of the cases (data available for 141 MPHP) (Figure 1 and Figure 2). For 152 MPHP, the SD-OCT B-scans passing through the plaque were available. RPE bulges or PED were detected in 98.7% of cases while pachyvessels were observed in 57.2% of areas with MPHP (Figure 2).

### 3.5. Evolution of MPHP

During a mean follow-up of 1.9 ± 1.4 years, another ICGA was available in 43 eyes. We found a stability of the number and size of MPHP in 33 eyes (76.7%). In 8 eyes (18.6%), we found a resolution of MPHP. In these eyes, RPE became atrophic, as shown in Appendix A. In 2 eyes, we observed the emergence of new MPHP.

## 4. Discussion

From 1965 onwards, a primitive RPE dysfunction has been suggested in the pathogenesis of CSCR, supported by the presence of the focal leaks from the RPE [1]. An involvement of the choroidal vasculature has been identified on ICG-A, that showed delayed choroidal arterial filling, choroidal venous dilatation and focal hyperfluorescent area [1,8,10,16]. According to Hayashi et al., as the delayed filling was associated with RPE degeneration, circulatory insufficiency has been proposed to cause the epitheliopathy [17]. The areas of ICG-A hyperfluorescence, called MPHP, seen on mid-phase ICG-A [18] at a time when the surrounding fluorescence decreases have been attributed to choroidal vessels permeability. Such regions of hyperfluorescence are reported in more than 90% of CSCR cases in most studies [10,19,20] and a risk site for fluorescein leakage and SRD [10,16,18]. In our cohort of CSCR patients, 62.4% of patients had a chronic form of the disease, and MPHP were found in almost 60% of eyes, suggesting that they are different from previously described area of hyperfluorescence and that they may characterize a specific type of patients.

In this study, MPHP were associated with the chronicity and with multiple signs suggestive of a chronic form of the disease such as irregular PED [15,21] and a subfoveal choroidal dilated vein potentially suggesting abnormal venous drainage and anastomosis [22,23]. In addition, when a focal leak was present, it was found within the plaque in 80% of the cases, in line with previous studies and suggesting that the RPE barrier could be weaker in these area.

Since MPHP were also observed in contralateral unaffected eyes, they could be a prognosis factor rather than a consequence of the chronicity, but further prospective studies are needed to confirm this assumption. Indeed, In our study, we found MPHP in 35.3% contralateral unaffected eyes and we observed the occurrence of subretinal detachment in one contralateral eye. However, a longer follow-up is mandatory to evaluate more accurately the incidence of subretinal detachment according to the presence of MPHP.

Interestingly, in areas with MPHP, pachyvessels were seen on early ICG-A in 42.8% of cases and on SD-OCT in 57.2% of cases while PED or RPE bulges were found in 98.7% of areas with MPHP. In other words, areas with MPHP were more frequently associated with abnormalities of the RPE layer on SD-OCT than with the presence of dilated veins. There was also a significant association between the areas of delayed ICG filling and the areas of MPHP, which could lead to hypoxic suffering of the RPE in these specific areas. Altogether, these observations indicate that MPHP occurred in areas of diseased RPE, even if 43% of areas with MPHP had a normal aspect on BAF, suggesting an “early” RPE dysfunction.

The mechanisms of increased fluorescence of ICG in the choroid and RPE after systemic injections are multiple. Since more than 95% of the systemically injected ICG binds to lipoproteins, ICG is assume to follow the transport mechanisms of these molecules in the choroid/RPE/retina complex [24]. The choriocapillaris is poorly permeable to circulating endogenous proteins despite its fenestration [25,26] that are highly VEGF-dependent [27]. In CSCR, no increase in VEGF levels have been measured [28,29] suggesting that increased passage through fenestrations is unlikely. Other mechanisms related to active transports rather than passive diffusion could be deregulated. The delayed visualization of ICG in MPHP (few minutes after ICG injection) correlates well with the active transports from vascular endothelial cells to the RPE cells, related to the Na+/K+ ATPase activity [12] or to caveolins [30], that have been shown to control the transcellular movement of albumin and lipoproteins from the vessels towards the RPE and photoreceptor cells [30]. Excessive passage of proteins in the outer retina through RPE would increase the interstitial pressure and explain the smokestack appearance of fluorescein leaking through the RPE following an oncotic gradient. Recently, Sakurada et al. [31] have reported an association between the presence of choroidal caverns and the presence of hyperfluorescent area on mid-phase ICG-A. Since caverns could correspond to lipoproteins and lipids, it could be assumed that ICG could also accumulate in these deposits [31], also reflecting abnormal transports between the choroid and the RPE since lipoproteins physiologically transfer beta carotene or xanthophylls from the choroid to the retina [32].

We assumed that MPHP could result from an excessive accumulation of ICG (and other macromolecules such as proteins and lipoproteins) either within the abnormal RPE, or in sub RPE deposits (Figure 3) and could thus be an early indicator of epitheliopathy, visible before any change is observed on BAF as autofluorescent changes were observed in only 57% of areas with MPHP. This finding is in line with cases in which MPHP tend to disappear when RPE hypoautofluorescence progresses after years of evolution, suggesting mainly a RPE loss (Appendix A). We assumed that in CSCR, MHPH could indicate an over transcellular transport of proteins or lipoproteins in diseased RPE.

The limitations of this study are its retrospective design and the fact that corresponding SD-OCT sections were not acquired for all MPHP areas. In addition, wide-field ICG-A was not performed and it could have provided a dynamic and more complete analysis of the choroidal circulation.

## 5. Conclusions

CSCR is a multifactorial disease with a complex pathogenesis and we propose to add an altered macromolecular transport from choroidal endothelial cells to the RPE/Bruch’s membrane, shown by the presence of MPHP on ICG-A, to the multiple pathogenic mechanisms. The presence of MPHP may indicate an “hyperporosity” of the RPE and/or Bruch’s membrane and be an early sign of epitheliopathy. Prospective longitudinal studies are needed to explore the prognosis value of MPHP in diseased and contralateral eyes of CSCR patients.

## Figures and Tables

**Figure 1 jcm-10-04525-f001:**
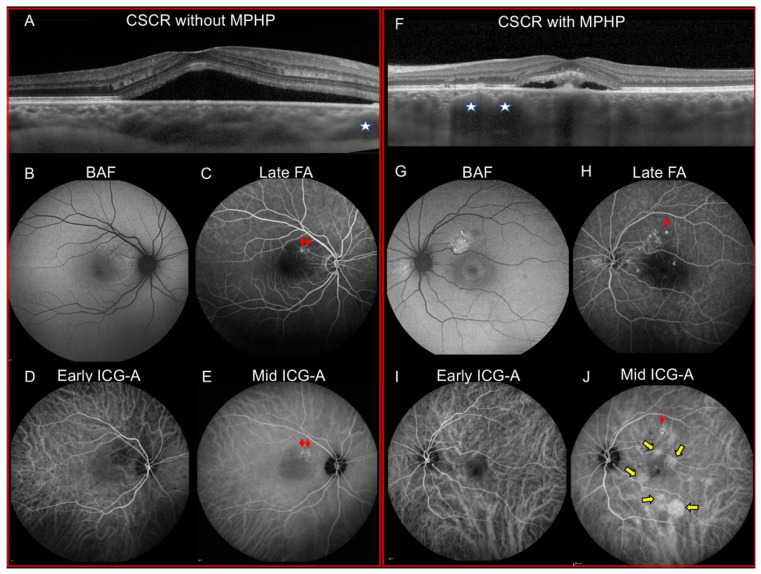
Multimodal imaging of patients with central serous chorioretinopathy (CSCR) without mid-phase hyperfluorescent plaques (MPHP) (**A**–**E**) and with MPHP (**F**–**J**). (**A**–**E**) A 35-year-old man with CSCR in the right eye. (**A**). The horizontal enhanced depth imaging (EDI) OCT centered on the fovea shows a macular serous retinal detachment (SRD). A dilated choroidal vessel is visualized (stars). (**B**). Blue-light fundus autofluorescence shows a mixed hyper- and hypo-autofluorescent area at the SRD. (**C**). Late-phase fluorescein angiography (FA) shows two focal leaks (arrows). (**D**). Early-phase indocyanine green angiography (ICG-A) is unremarkable. (**E**). Mid-phase ICG-A shows the two focal leaks (arrows) visualized on FA without MPHP. (**F**–**J**) A 46-year-old man with CSCR in the left eye. (**F**). An EDI-OCT centered on the fovea shows a macular SRD associated with dilated choroidal vessels (stars). (**G**). Blue-light fundus autofluorescence shows a superior hyper-autofluorescent area consistent with a previous SRD. (**H**). Late-phase FA shows a leaking point (red arrow) and an ill-defined hyperfluorescent area consistent with retinal pigment epithelium changes. (**I**). Early-phase ICG-A shows inferior dilated choroidal veins. (**J**). Mid-phase ICG-A shows multifocal MPHP (yellow arrows). The focal leak seen on FA is also detected on mid-phase ICG-A (red arrow).

**Figure 2 jcm-10-04525-f002:**
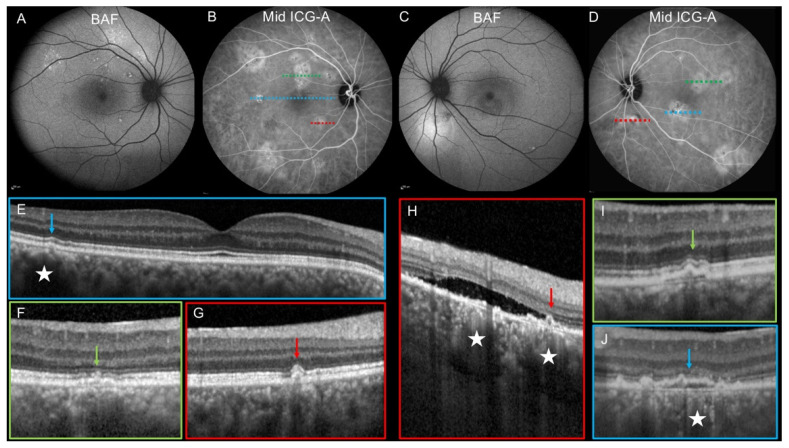
Multimodal imaging of a 37-year-old man with bilateral chronic central serous chorioretinopathy. (**A**,**B**,**E**–**G**). Right eye. (**A**). Blue-light fundus autofluorescence (BAF) shows a superior hyper-autofluorescent area consistent with previous serous retinal detachments. (**B**). Mid-phase indocyanine green angiography (ICG-A) shows multifocal mid-phase hyperfluorescent plaques (MPHP) without retinal pigment epithelium (RPE) damages detected on BAF. (**E**). The horizontal OCT B-scan passing through the fovea shows a RPE bulge at the level of the temporal MPHP (arrow). (**F**,**G**). The horizontal OCT B-scan passing through the MPHP (green and red dotted lines) shows an irregular pigment epithelium detachment (PED, (**F**), green arrow) and a RPE bulge ((**G**), arrow). (**C**,**D**,**H**–**J**). Left eye. (**C**). BAF shows a hyper-autofluorescent area inferior to the optic disc that corresponds to a subretinal detachment (SRD, (**H**)). (**D**). Mid-phase ICG-A shows multifocal MPHP. (**H**) The horizontal OCT B-scan passing through the MPHP (red dotted-line) shows a serous retinal detachment associated with RPE irregularities (red arrow) and dilated choroidal vessels (stars). (**I**,**J**). The horizontal OCT B-scan passing through the MPHP (green and blue dotted-line) shows small irregular PED (green arrow and blue arrow). A dilated choroidal vessel is visualized behind the PED (star, (**J**)).

**Figure 3 jcm-10-04525-f003:**
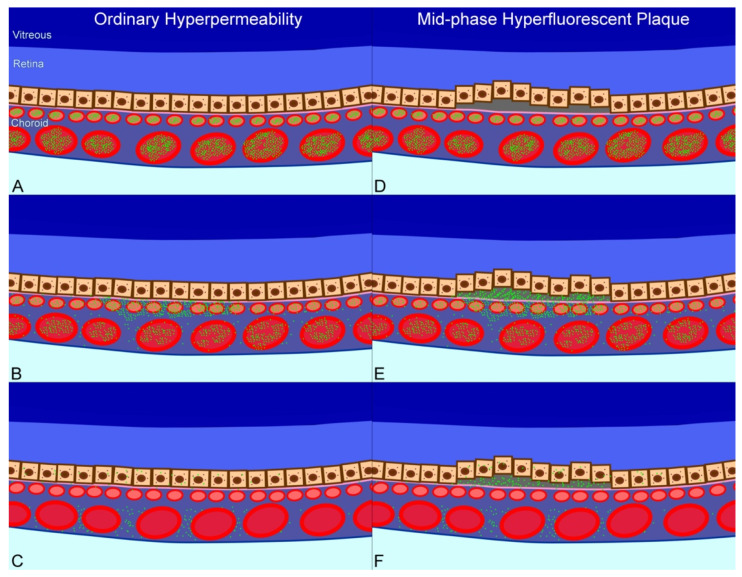
Manifestations of hyperpermeability in central serous chorioretinopathy. (**A**). Soon after injection of indocyanine green, the intravascular dye concentration is high. (**B**). In areas of choroidal vascular hyperpermeability there is leakage of dye into the choroidal stroma. (**C**). The dye is removed from the blood stream by the liver. The dye in the stroma may diffuse back into the blood or spread posteriorly between the larger choroidal vessels. This produces a silhouetting of the large choroidal vessels against a hyperfluorescent background. (**D**). In areas with mid-phase hyperfluorescent plaques (MPHP) there appears to be localized, shallow elevations of the RPE monolayer. In some cases, the space has reflectivity greater and more heterogeneous than what would be expected from simple fluid alone. (**E**). Dye leaking in areas of chorodal vascular hyperpermeability may diffuse into the choroidal stroma or into the sub-retinal pigment epithelial space. The fluorescence from the anteriorly located dye masks the visible details of the underlying choroid. (**F**). In the later portions of the mid-phase, there is retention of dye in the sub-RPE space. The contrast between the regions with retained dye and the lesser amounts remaining in the choroid produces the plaque-like appearance.

**Table 1 jcm-10-04525-t001:** Demographics, clinical and imaging data of eyes with central serous chorioretinopathy (CSCR) included in the study.

Characteristics	Result (133 Eyes of 100 Patients)
**Demographic data**	
Age, mean ± SD, year	47.6 ± 10.2
Gender, males, *n* (%)	91 (91%)
Corticosteroid intake ^a^ *n* (%)	42 (47.2%)
Shift work ^b^ *n* (%)	29 (35.4%)
**Previous CSCR treatments ^c^**	
Laser Photocoagulation *n* eyes (%)	20 (16.7%)
PDT, *n* eyes (%)	11 (9.2%)
**Clinical form of CSCR**	
Acute/recurrent CSCR, *n* (%)	50 (37.6%)
Chronic CSCR	83 (62.4%)
**Clinical data**	
BCVA, LogMAR, mean ± SD (Snellen)	0.23 ± 0.36 (20/32)
**OCT findings**	
Subfoveal choroidal thickness, mean ± SD (μm)	461.3 ± 104.5
PED and/or RPE bulge, *n* (%)	117 (88.7%)
At least one dome-shaped PED, *n* (%)	42 (35.9%)
At least one irregular PED, *n* (%)	50 (37.6%)
**Blue-light fundus Autofluorescence findings**	
Hyper/hypoautofluorescent area, *n* (%)	121 (91%)
Gravitational tracks, *n* (%)	36 (27.1%)
**Fluorescein Angiography findings**	
RPE changes, *n* (%)	114 (85.7%)
CSC ink-blot or smokestack leakage, *n* (%)	50 (37.6%)
**Indocyanine green Angiography findings**	
Pachyvessels, *n* (%)	71 (53.4%)
Foveal pachyvessel, *n* (%)	19 (14%)
Mid-phase hyperfluorescent plaque, *n* eyes (%)	79 (59.4%)

SD: Standard Deviation; PDT: Photodynamic Therapy; BCVA: Best-Corrected Visual Acuity; OCT: Optical Coherence Tomography; RPE: retinal pigment epithelium; PED: pigment epithelium detachment. ^a^ data available for 89 patients, ^b^ data available for 82 patients, ^c^ data available for 120 eyes.

**Table 2 jcm-10-04525-t002:** Comparison between Eyes with and without Mid-phase Hyperfluorescent Plaques on Indocyanine Green Angiography.

Characteristics	CSCR with Mid-Phase Hyperfluorescent Plaques (*n* = 61 Patients, 79 Eyes)	CSCR without Mid-Phase Hyperfluorescent Plaques (*n* = 39 Patients, 54 Eyes)	*p* Value
**Demographics**			
Age, mean ± SD, year	48.1 ± 11	46.9 ± 8.8	0.53 ^a^
Gender, males (%)	58 (95.1%)	33 (84.6%)	0.15 ^b^
Corticosteroid intake *, *n* (%)	30 (54.5%)	12 (35.3%)	0.08 ^c^
Shift work **, *n* (%)	21 (39.6%)	8 (27.6%)	0.28 ^c^
**Clinical form of CSCR**, *n* (%)			
Acute/recurrent CSCR	13 (16.5%)	37 (68.5%)	<0.001 ^c^
Chronic CSCR	66 (83.5%)	17 (31.5%)	
**Clinical data**			
BCVA (LogMAR), mean ± SD	0.28 ± 0.4	0.17 ± 0.29	0.25 ^a^
**OCT findings**			
Subfoveal choroidal thickness, mean ± SD (μm)	463.8 ± 100.4	457.9 ± 111	0.6 ^a^
PED and/or RPE bulge, *n* (%)	71 (89.9%)	47 (87%)	0.61 ^c^
At least one dome-shaped PED, *n* (%)	23 (29.1%)	19 (35.2%)	0.46 ^c^
A least one irregular PED, *n* (%)	40 (51%)	10 (18.5%)	<0.001 ^c^
**Autofluorescence findings**			
Hyper/hypoautofluorescent area, *n* (%)	79 (100%)	42 (78.8%)	<0.001 ^c^
Gravitational tracks, *n* (%)	30 (38%)	6 (11.3%)	0.001 ^c^
**Fluorescein Angiography findings**			
RPE changes, *n* (%)	79 (100%)	35 (64.8%)	<0.001 ^c^
CSCR ink-blot or smokestack leakage, *n* (%)	25 (31.7%)	25 (46.3%)	0.09 ^c^
**Indocyanine Green Angixography findings**			
Pachyvessels, *n* (%)	47 (59.5%)	24 (44.5%)	0.09 ^c^
Foveal pachyvessel, *n* (%)	18 (22.8%)	1 (1.9%)	0.001 ^c^

BCVA: Best-Corrected Visual Acuity; CSCR: central serous chorioretinopathy; OCT: Optical Coherence Tomography; PED: pigment epithelium detachment; RPE: retinal pigment epithelium; SD: Standard Deviation. ^a^ Mann-Whitney test; ^b^ Chi-square test with Yates continuity; ^c^ Chi-square. * data available for 89 patients, ** data available for 82 patients.

**Table 3 jcm-10-04525-t003:** Multiple Logistic Regression Analysis of factors associated with the presence of mid-phase hyperfluorescent plaques on indocyanine angiography in patients with central serous chorioretinopathy.

	Multivariate Analysis	
Variable	OR (95%, CI)	*p* Value
Chronic CSCR	3.5 (1.28–9.74)	0.015
Irregular PED	3 (1.2–7.5)	0.018
RPE changes on fluorescein angiography	5.8 (1.9–17.7)	0.002
Foveal pachyvessel on indocyanine green angiography	4.3 (1.1–16.6)	0.036

CSCR: central serous chorioretinopathy; CI: confidence interval; OR: odds ratio; RPE: retinal pigment epithelium; PED: pigment epithelium detachment.

**Table 4 jcm-10-04525-t004:** Multimodal analysis of the areas with mid-phase hyperfluorescent plaques (MPHP) on indocyanine green angiography in patients with central serous chorioretinopathy.

Variable	Results
**Number of MPHP analyzed (*n*)**	249 areas with MPHP area in 79 eyes
**Indocyanine Green Angiography findings**	
Delayed choroidal filling in areas with MPHP, %	77.3%
Dilated choroidal vein in areas with MPHP, %	42.8%
Hypofluorescent spots in the late phase in areas with MPHP, %	63.2%
**Fluorescein angiography findings**	
RPE window defect in areas with MPHP, %	73.1%
**Blue-light fundus autofluorescence findings** Changes in areas with MPHP, %	57.3%
**SD-OCT findings**	
Pigmented Epithelium Detachment/bulges of the RPE in areas with MPHP, %	98.7%
Pachyvessels in areas with MPHP, %	57.2%
Serous Retinal Detachment in areas with MPHP, %	41.1%

MPHP: mid-phases hyperfluorescent plaques; SD-OCT: spectral domain optical-coherence tomography; RPE: retinal pigment epithelium.

## Data Availability

Data available on request from the corresponding author. The data are not public available due to privacy and ethical.

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
