# Peer review of "Mid-Phase Hyperfluorescent Plaques Seen on Indocyanine Green Angiography in Patients with Central Serous Chorioretinopathy"

_jcm, 2021, doi:10.3390/jcm10194525_

Round 1

Reviewer 1 Report

I reviewed the manuscript entitled “Mid-phase hyperfluorescent plaques seen on indocyanine green angiography in patients with central serous chorioretinopathy”by Bousquet et al.

In this paper, the Authors assessed the presence of mid-phase hyperfluorescent plaques (MPHP) as early sign of epitheliopathy in central serous chorioretinopathy (CSCR) patients.

This paper was well-written, with fluent English. The iconography was good.

However, some considerations should be made:

  • What was the time between the diagnosis of CSCR and the ICGA execution? Could the author could report how these MPHP have changed over time?
  • What was the aspect of these areas in the late stages of the angiogram?
  • Has the author ever seen a single intermediate phase hyperfluorescent plaque corresponding to a late plaque with underlying macular neovascularization (MNV)? The MNV could be associated with FI-PED or other RPE abnormalities and I think it is important to highlight the differences between RPE abnormalities of CSCR epitheliopathy and MNV imaged in ICGA. Hence, I suggest adding a figure to differentiate ICGA plaque with and without MNV in CSCR patients. This could help the reader focus on MPHP as an early sign of RPE abnormalities and not get confused about the presence of MNV.
  • In how many patients was this finding also found in the contralateral unaffected eye? In this case, can the author provide the time of onset of CSCR in the fellow eye?

Author Response

I reviewed the manuscript entitled “Mid-phase hyperfluorescent plaques seen on indocyanine green angiography in patients with central serous chorioretinopathy” by Bousquet et al.

In this paper, the Authors assessed the presence of mid-phase hyperfluorescent plaques (MPHP) as early sign of epitheliopathy in central serous chorioretinopathy (CSCR) patients.

This paper was well-written, with fluent English. The iconography was good.

We thank the reviewer for these encouraging comments.

However, some considerations should be made:

  • What was the time between the diagnosis of CSCR and the ICGA execution?

In our center, most of patients underwent a fluorescein and indocyanine angiography in the first month even in acute form of the disease. However, in some cases the ICGA was performed at 3 months in cases of persistent serous retinal detachment to guide a photodynamic therapy treatment. In clinical practice, the indication of ICG-A depends on the physicians who examined the patient.

We agree that ICG-A is more often realized in chronic, long lasting CSCR that is why we included a high proportion of chronic CSCR in our study. Another reason is that the patients included in this study came from a tertiary center.

  • Could the author could report how these MPHP have changed over time?

Thank you for this interesting question. In the revised version of the manuscript, we added a paragraph on the evolution of the MPHP in the result section :

“During a mean follow-up of 1.9 ± 1.4 years, another ICGA was available in 43 eyes. We found a stability of the number and the size of MPHP in 33 eyes (76.7%). In 8 eyes (18.6%), we found a resolution of MPHP. In these eyes RPE became atrophic, as shown in Suppl Fig S2. In 2 eyes, we observed the emergence of new MPHP.” (line 242-245).

Interestingly, our results are in agreement with Piccolino et al’s study [1], which showed that the plaques remained unchanged even when the subretinal fluid resolved and until the end of the follow-up period, which was 2 years in their study.

  • What was the aspect of these areas in the late stages of the angiogram?

As described line 46, the characteristic of the MPHPs is that they fade, becoming isofluorescent at the late phase of the ICGA and, with hypofluorescent spots in 63.2% of cases as shown in Table 4.

In the revised version of our manuscript, we have added a new supplementary figure to show the late phase of MPHP (supplementary Figure 1 A-C).

  • Has the author ever seen a single intermediate phase hyperfluorescent plaque corresponding to a late plaque with underlying macular neovascularization (MNV)? The MNV could be associated with FI-PED or other RPE abnormalities and I think it is important to highlight the differences between RPE abnormalities of CSCR epitheliopathy and MNV imaged in ICGA. Hence, I suggest adding a figure to differentiate ICGA plaque with and without MNV in CSCR patients. This could help the reader focus on MPHP as an early sign of RPE abnormalities and not get confused about the presence of MNV.

We thank the reviewer for this very interesting point.

To avoid any confusions with MNV, we have replaced the term FIPED with “irregular PED” in the revised manuscript.

In most of cases, MPHP were extramacular as shown in Fig 1-2-S1-S2 and these MPHP were not imaged with OCT-angiography.

However, the main difference between the MPHP and the hyperfluorescent area of a MNV is the increase hyperfluorescence during the late-phase of ICG-A in case of MNV.

We added a new supplementary Figure S1, showing ICGA during mid-phase and late-phase of a CSCR patient with MPHP and a CSCR patient with MNV to highlight the difference between both hyperfluorescences.

  • In how many patients was this finding also found in the contralateral unaffected eye? In this case, can the author provide the time of onset of CSCR in the fellow eye

In our study, 68 patients had a unilateral CSCR and 32 patients had CSCR in both eyes.

We have added this result line 154 : “A unilateral form of CSCR was observed in 68 patients while 32 patients had both eyes affected.”

Among 68 patients with unilateral CSCR, we found MPHP in 24 (35.3%) contralateral unaffected eyes. The mean follow-up of these unilateral CSCR with MPHP in the contralateral eye was 1.1 ± 1.5 years and we observed the occurrence of subretinal detachment in only one contralateral eye. In this fellow eye, the CSCR occurred 2.7 years after the diagnosis of CSCR in the first eye. We can hypothesize that a higher rate of SRD would have been observed in the contralateral unaffected eye with a longer follow-up.

References :

  1. Piccolino, F.C.; Borgia, L.; Zinicola, E.; Zingirian, M. Indocyanine Green Angiographic Findings in Central Serous Chorioretinopathy. Eye (Lond) 1995, 9 ( Pt 3), 324–332, doi:10.1038/eye.1995.63.

Reviewer 2 Report

the manuscript is well structured, the topic is very current and the study with indocyanine angiography is conducted in a very specific way.
I can only congratulate the authors also for the way in which the data is presented, which appears very linear and clear. reading is fluid and clean. I only recommend some changes that the authors can make to make the manuscript even more publishable.
I recommend extending the introductory part. the authors could increase the part on the etiopathogenesis of central serous chorioreti
nopathy.

In addition, in this regard, I advise the authors to dwell in some recent studies on etiopathogenesis (psychophysics).
If the authors deem it necessary and useful, I recommend inserting some bibliographic notes both in the text of the manuscript and in the bibliographic notes (annexed to note 24-23 in line 267)

"Genovese G, Meduri A, Muscatello MRA, Gangemi S, Cedro C, Bruno A, Aragona P, Pandolfo G. Central Serous Chorioretinopathy and Personality Characteristics: A Systematic Review of Scientific Evidence over the Last 10 Years (2010 to 2020). Medicina (Kaunas). 2021 Jun 16;57(6):628. doi: 10.3390/medicina57060628. PMID: 34208694; PMCID: PMC8235071.

Pandolfo G, Genovese G, Bruno A, Palumbo D, Poli U, Gangemi S, Aragona P, Meduri A. Sharing the Same Perspective. Mental Disorders and Central Serous Chorioretinopathy: A Systematic Review of Evidence from 2010 to 2020. Biomedicines. 2021 Aug 23;9(8):1067. doi: 10.3390/biomedicines9081067. PMID: 34440271.

I recommend a minimal correction of the English language.

Author Response

the manuscript is well structured, the topic is very current and the study with indocyanine angiography is conducted in a very specific way.
I can only congratulate the authors also for the way in which the data is presented, which appears very linear and clear. reading is fluid and clean. I only recommend some changes that the authors can make to make the manuscript even more publishable.

We thank the reviewer for these positive comments.

I recommend extending the introductory part. the authors could increase the part on the etiopathogenesis of central serous chorioreti
nopathy.

In addition, in this regard, I advise the authors to dwell in some recent studies on etiopathogenesis (psychophysics).
If the authors deem it necessary and useful, I recommend inserting some bibliographic notes both in the text of the manuscript and in the bibliographic notes (annexed to note 24-23 in line 267)

"Genovese G, Meduri A, Muscatello MRA, Gangemi S, Cedro C, Bruno A, Aragona P, Pandolfo G. Central Serous Chorioretinopathy and Personality Characteristics: A Systematic Review of Scientific Evidence over the Last 10 Years (2010 to 2020). Medicina (Kaunas). 2021 Jun 16;57(6):628. doi: 10.3390/medicina57060628. PMID: 34208694; PMCID: PMC8235071.

Pandolfo G, Genovese G, Bruno A, Palumbo D, Poli U, Gangemi S, Aragona P, Meduri A. Sharing the Same Perspective. Mental Disorders and Central Serous Chorioretinopathy: A Systematic Review of Evidence from 2010 to 2020. Biomedicines. 2021 Aug 23;9(8):1067. doi: 10.3390/biomedicines9081067. PMID: 34440271.

We have now added in the revised manuscript the following paragraph on the risk factors of CSCR :

“To date, several risk factors for CSCR have been identified. The most consistent risk factor is an exposure to glucocorticoids[3]. Several other predisposing factors or contributing factors have been identified as: psychological stress, anxiety and maladaptive personality[4,5], hypertension and cardiovascular disease, sleep disturbance and shift work[6], allergic disorders, pregnancy, peptic ulcer disease and also genetic risk factors.[7]” (line 43-48)

I recommend a minimal correction of the English language.

The manuscript was edited by Sophie Pegorier, a medical and scientific professional translator.
